# Suppression of the Proliferation of Huh7 Hepatoma Cells Involving the Downregulation of Mutant p53 Protein and Inactivation of the STAT 3 Pathway with Ailanthoidol

**DOI:** 10.3390/ijms23095102

**Published:** 2022-05-04

**Authors:** Tsui-Hwa Tseng, Chau-Jong Wang, Yean-Jang Lee, Yi-Chia Shao, Chien-Heng Shen, Ko-Chao Lee, Shui-Yi Tung, Hsing-Chun Kuo

**Affiliations:** 1Department of Medical Applied Chemistry, Chung Shan Medical University, Taichung 40201, Taiwan; tht@csmu.edu.tw (T.-H.T.); a14253681@gmail.com (Y.-C.S.); 2Department of Medical Education, Chung Shan Medical University Hospital, Taichung 40201, Taiwan; 3Department of Health Diet and Industry Management, Chung Shan Medical University, Taichung 40201, Taiwan; wcj@csmu.edu.tw; 4Department of Medical Research, Chung Shan Medical University Hospital, Taichung 40201, Taiwan; 5Department of Chemistry, National Changhua University of Education, Changhua 50007, Taiwan; leeyj@cc.ncue.edu.tw; 6Department of Hepato-Gastroenterology, Chang Gung Memorial Hospital, Chiayi 61363, Taiwan; gi2216@adm.cghmh.org.tw; 7Division of Colorectal Surgery, Department of Surgery, Chang Gung Memorial Hospital, Kaohsiung Medical Center, Chang Gung University College of Medicine, Kaohsiung 83301, Taiwan; kclee@cgmh.org.tw; 8Graduate Institute of Clinical Medical Sciences, College of Medicine, Chang Gung University, Chiayi 61363, Taiwan; 9Division of Basic Medical Sciences, Department of Nursing, Chang Gung University of Science and Technology, Chiayi 61363, Taiwan; 10Research Fellow, Chang Gung Memorial Hospital, Chiayi 61363, Taiwan; 11Research Center for Food and Cosmetic Safety, College of Human Ecology, Chang Gung University of Science and Technology, Taoyuan 33303, Taiwan; 12Chronic Diseases and Health Promotion Research Center, Chang Gung University of Science and Technology, Chiayi 61363, Taiwan

**Keywords:** ailanthoidol, hepatoma, mutant *p53*, STAT3, apoptosis, cell cycle

## Abstract

Ailanthoidol (ATD) has been isolated from the barks of *Zanthoxylum ailanthoides* and displays anti-inflammatory, antioxidant, antiadipogenic, and antitumor promotion activities. Recently, we found that ATD suppressed TGF-β1-induced migration and invasion of HepG2 cells. In this report, we found that ATD exhibited more potent cytotoxicity in Huh7 hepatoma cells (mutant *p53*: Y220C) than in HepG2 cells (wild-type *p53*). A trypan blue dye exclusion assay and colony assay showed ATD inhibited the growth of Huh7 cells. ATD also induced G1 arrest and reduced the expression of cyclin D1 and CDK2. Flow cytometry analysis with Annexin-V/PI staining demonstrated that ATD induced significant apoptosis in Huh7 cells. Moreover, ATD increased the expression of cleaved PARP and Bax and decreased the expression of procaspase 3/8 and Bcl-xL/Bcl-2. In addition, ATD decreased the expression of mutant p53 protein (mut*p53*), which is associated with cell proliferation with the exploration of p53 siRNA transfection. Furthermore, ATD suppressed the phosphorylation of the signal transducer and activator of transcription 3 (STAT3) and the expression of mevalonate kinase (MVK). Consistent with ATD, the administration of S3I201 (STAT 3 inhibitor) reduced the expression of Bcl-2/Bcl-xL, cyclin D1, mutp53, and MVK. These results demonstrated ATD’s selectivity against mut*p53* hepatoma cells involving the downregulation of mut*p53* and inactivation of STAT3.

## 1. Introduction

Hepatocellular carcinoma (HCC), the most common primary malignant tumor in liver cancer cases, is a complex disease caused by a variety of risk factors. Conventional types of liver cancer treatment, including surgical resection, radiotherapy, and chemotherapy, have been either limited in application or ineffective [1]. Transplantation of the liver is believed to be the only viable treatment; however, it is not easy to find the proper donor. Although scientists have generated intense research efforts to explore cellular, molecular, and physiological mechanisms of the disease for developing prevention and therapy strategies [2], the mortality rate of HCC remains high.

The transcription factor p53 is activated in response to various stresses including nutrient deprivation, DNA damage, oncogene activation, and hypoxia. p53 is a well-established tumor suppressor and guardian of the genome that induces apoptosis and cell cycle arrest by activating downstream target genes [3]. However, *p53* is mutated in around half of all human cancers. It is generally believed that p53 loses its tumor suppressor function because of a mutation in *p53*. Certain types of *p53* mutations are gain-of-function mutations, which have been shown to have oncogenic functions [4]. HCC is a lethal malignancy associated with poor prognosis and a high recurrence. Effective HCC therapeutics still await a molecular understanding of the mechanisms promoting the development of selective and precise agents. HCC has a high rate of mutation in tumor suppressor protein p53, leading to the loss of its tumor suppressor activity and, in certain cases, gain-of-function activities that promote cell proliferation, tumor progression, and drug resistance [5]. Thus, mutant *p53* has become an important target for the development of anticancer agents in HCC.

The signal transducer and activator of transcription 3 (STAT3) is a pivotal transcriptional factor of multiple promoting genes in cancer development and immune evasion [6]. Phosphorylated STAT3s dimerize each other and translocate into the nucleus before activating the downstream genes. Under a normal physiological state, STAT3 activation is usually transient in the continuous stimulation of cytokines and contributes to protecting normal hepatocytes from inflammatory insults. It has been reported that constitutive phosphorylation of STAT3 in tumor tissue is correlated with poor prognosis in HCC patients [7]. Thereafter, the inactivation of the STAT3 signal pathway is a promising strategy in anti-HCC treatment.

Ailanthoidol (ATD), a neolignan, has been isolated from the bark of *Zanthoxylum ailanthoides* (Rutaceae), of which the dried fruit is used as a spice in Taiwan. Our previous study demonstrated that ATD displays antitumor promotion effects using the multistep skin cancer model induced by 12-o-tetradecanoylphobol-13-acetate [8]. Recently, we found that ATD suppresses TGF-β1-promoted migration and invasion in HepG2 cells [9]. Kim and Jun reported that ATD has in vitro and in vivo anti-inflammatory effects [10]. Park et al. found that ATD possesses antiadipogenic activities [11]. In addition, ATD is a benzofuran derivative and indicates diverse pharmacological activities, including anticancer activities [12]. As the anticancer properties of ATD have not been well clarified, this study investigated the antiproliferation effects and molecular mechanism of ATD in hepatoma cells.

## 2. Results

### 2.1. Effects of ATD on the Growth of Huh7 and HepG2 Cells

To understand cell viability under ATD treatment on the hepatocellular carcinoma cells (HCCs), a range of concentrations (0–80 μM) was evaluated in the Huh7 and HepG2 cells with an MTT assay. As shown in Figure 1, ATD suppressed cell viability in Huh7 cells, with IC50 values of 45 μM and 22 μM at 24 h and 48 h, respectively, while the IC50 value in HepG2 cells was above 80 μM. In addition, to examine the effects of ATD on the growth of Huh7 cells, a trypan blue dye exclusion assay and colony assay were performed. The results demonstrated that ATD decreased the growth of Huh7 cells in a time-dependent manner (Figure 2A). Furthermore, a colony formation assay confirmed that ATD decreased the growth of Huh7 cells significantly (Figure 2B). Thereafter, we proceeded to study the antitumor potential and mode of action of ATD in Huh7 cells.

### 2.2. Effect of ATD on the Cell Cycle Distribution of Huh7 Cells

To determine the cellular mechanism preventing cancer cell proliferation, we examined cell cycle profiles in Huh7 cells with or without ATD administration at various times, using flow cytometry. When the cells were administrated with ATD (10 μM), the proportion of the subG1 and G0/G1 phases tended to significantly increase, compared with the control (0 h), while the G2/M phase was decreased (Figure 3A,B). In addition, ATD downregulated the expression levels of the checkpoint proteins involved in the regulation of G1 phase transition, such as cyclin D1 and CDK2 (Figure 3C).

### 2.3. Induction Apoptosis by ATD in Huh7 Cells

To examine the mechanism of ATD-induced cytotoxicity, apoptotic induction of ATD (10, 20, and 40 μM) was evaluated by flow cytometry analysis with Annexin V/PI double staining. While the percentage of early apoptotic cells in the control group was 3%, in the ATD-treated groups, it increased from 4.13% to 13.43% (Figure 4A). ATD significantly increased the percentage of total apoptotic cells (early plus late) in a dose-dependent manner, from 8.35% to 22.49%, while in the control group, it was 5.95% for Huh7 cells (Figure 4A,B). To further characterize the cell death process, we investigated the downstream expression of apoptotic associated proteins using a Western blot assay. ATD decreased the expression levels of procaspase 3, procaspase 8, Bcl-xL, and Bcl-2 but increased the levels of Bax and cleavage poly(ADP-ribose) polymerase (PARP) (Figure 4C).

### 2.4. Induction Apoptosis by ATD in Huh7 Cells

As ATD exhibited a marked reduction in the IC50 value in Huh7 cells (mutant *p53* Y220C), compared with HepG2 cells (wild-type *p53*), the effect of ATD on the p53 expression in Huh7 cells could be determined. The immunoblotting assay against the p53 antibody (DO-1), which is recommended for detection of wide-type and mutant *p53*, revealed that ATD reduced the expression of p53 in Huh7 cells in a dose-dependent manner (Figure 5A). In addition, according to the immunofluorescence analysis against the p53 antibody (PAb240), which is recommended for mutant *p53* under non-denaturing conditions, ATD reduced the fluorescence of p53 in Huh7 and PLC/PRF/5 cells (mutant p53 R249S), compared with the positive control group, respectively, while the negative control of HepG2 (wild-type *p53*) did not exhibit green fluorescence (Figure 5B). In order to determine whether mutant *p53* was involved in the ATD-induced antiproliferation, we conducted a p53 knockdown experiment using the transfection of *p53* siRNA. Although the cellular levels of p53 in Huh7 cells transfected with p53 siRNA were not completely knocked down, a distinct downregulation of the cellular p53 levels was observed (Figure 6A). The CCK-8 assay indicated that p53 knockdown indeed decreased cell viability (Figure 6B). In addition, ATD treatment significantly enhanced the antiproliferation property in the p53 knockdown cells (Figure 6B), indicating that mutant p53 was involved in ATD-induced apoptosis and cell cycle arrest in Huh7 cells.

### 2.5. ATD-Induced Antiproliferation of Huh7 Cells by Suppressing the STAT3 Pathway

STAT3 has recently emerged as a potential therapeutic target for HCC [7]. In addition, it has been demonstrated that STAT3 may sustain mut*p53* levels due to its interplay with the mevalonate pathway, which increases its stability [13]. Thereafter, we determined the effect of ATD on the phosphorylation of STAT3 and the expression of mevalonate kinases (MVK), a downstream target gene product of the STAT3 pathway. As shown in Figure 7A, ATD decreased the level of phosphorylated STAT3 and MVK. Consistent with ATD, S3I201, an inhibitor of STAT3, decreased the expression of Bcl-xL/Bcl2, p53, and MVK (Figure 7B,C). For one other cell line, PLC/PRF/5 cells (mut*p53* R249S), ATD decreased expression of p53 (DO-1), MVK and Bcl-2, Bcl-XL, cyclin D1 as well as and phosphorylation of Stat3 (Figure 7D,E).

## 3. Discussion

HCC, which accounts for nearly 80% of all liver cancer cases, is a heterogeneous type of cancer caused by a variety of risk factors, including exposure to the hepatitis virus, food contaminated with Aflatoxin B1, heavy alcohol intake, and obesity [14,15]. The incidence of HCC is rising rapidly worldwide. In addition, since HCC is diagnosed at a late stage in most cases, surgical resection and liver transplantation are not practical therapy options. Metastasis and recurrence are quite common. Therefore, the development of a promising compound with target therapy potential is an urgent task. Plants are major food and pharmaceutical sources for humans. Some phytochemicals, such as alkaloids, diterpenoids, and sesquiterpenes, display therapeutic potential for cancer treatment [16]. However, these therapeutic phytochemicals are also associated with adverse side effects, such as cardiovascular diseases, vomiting, renal dysfunction, and myelotoxicity. Thereafter, scientists have dedicated themselves to developing phytochemicals with minimal side effects and good bioavailability. Lignans and neolignans may possess great potential for anticancer treatment and display good safety profiles [12,17,18]. In the present study, ATD, a neolignan isolated from the bark of *Zanthoxylum ailanthoides* [19], exhibited antiproliferation potential in Huh7 hepatoma cells, which was related to the induction of cell cycle arrest and the activation of apoptosis. Cell cycle arrest was mediated by the ATD-induced cyclin D1 and CDK2 expression, while apoptosis was activated by ATD-downregulated Bcl-xL/Bcl2 and augmented Bax, resulting in the activation of caspase 3. For a real application, animal studies of ATD are required in the future.

The tumor suppressor p53 regulates the transcription of numerous downstream target genes involved in cell cycle arrest, apoptosis, and metabolism. Loss of p53 activity by gene deletion or mutations in normal cells causes uncontrolled cell proliferation, leading to immortalization and, ultimately, cancer. Additionally, mutant *p53* shows oncogenic gain-of-function activities, such as enhanced tumor progression, metastasis potential, and drug resistance [20]. As a result, obtaining efficient inhibitors against mutant *p53* cancer cells remains an urgent task for medicine development. Reactivation of the wild-type *p53* function and expression or abrogation of mutant *p53* protein may halt cancer progression [21]. Accumulation of mutant *p53* is critical for the gain of function related to *p53* mutation, including enhanced cell growth and tumor progression; however, the manner in which mut*p53* is regulated and promotes cancer progression is not well understood [4]. Enzymes controlling p53 proteasomal degradation or stability and some microRNA have been considered to regulate mutant *p53* levels [13,22]. In the present study, we found that ATD had more potent cytotoxicity in Huh7 cells (mutant *p53*) than in HepG2 cells (WT *p53*), which was associated with reducing the level of mut*p53*. According to our results, ATD blocked the STAT3 pathway and mediated the abrogation of mut*p53*. Whether ATD affects the miRNA or enzymes associated with proteasomal degradation requires further clarification. Our data implicated that ATD displayed potent anticancer potential in mut*p53*-based HCC by impairing the gain of function of mutant *p53*.

Among the diverse signaling molecules, STAT3 is considered an oncogenic factor in HCC [7]. Under a normal physiological state, STAT3 activation is usually transient, even in the continuous stimulation of cytokines, and contributes to protecting normal hepatocytes from inflammatory and toxic insults. In HCC, the persistent activation of STAT3 changes the gene transcriptions associated with cell survival, proliferation, invasion, and angiogenesis. The pro-proliferative role of STAT3 is related to its antiapoptotic functions toward HCC via upregulating antiapoptotic proteins such as Bcl-xL. Furthermore, constitutive phosphorylation of STAT3 in tumor tissue is closely correlated with a poor prognosis in HCC patients [6]. Recently, it has been reported that STAT3 sustains mut*p53* expression due to its interplay with the mevalonate pathway, which increases the stability of mut*p53* and prevents its degradation from proteasome [13]. In the present study, ATD inhibited the p-STAT3, MVK, and mut*p53* levels in Huh7 cells. According to Figure 7, we supposed that ATD blocked the STAT3 pathway mediating a reduction in mut*p53* protein in Huh7 cells. Although we found that ATD reduced the level of MVK (the downstream target gene product of STAT3, which might affect the stability of mut*p53*), the real interplay between STAT3 and mut*p53* needs further elucidation. In addition to reducing the gain-of-function activity of mut*p53*, ATD also triggered apoptosis by decreasing the expression of Bcl-xL and Bcl-2, which is associated with the inactivation of the STAT3 pathway. Additional studies are still needed to elucidate the action mechanisms of the ailanthoidol (ATD) as a chemopreventive and therapeutic agent in in vivo xenograft mouse models.

## 4. Materials and Methods

### 4.1. Materials

Dulbecco’s modified Eagle’s medium (DMEM), phosphate-buffered saline (PBS), fetal bovine serum (FBS), penicillin–streptomycin–neomycin (PSN), and trypsin–EDTA were purchased from Gibco Ltd. (Grand Island, NY, USA). Primary antibodies against p53(DO-1), p53(Pab-240), CDK2, Bax, Bcl-2, Bcl-xL pro-caspase 3/8, STAT3, MVK, GADPH, and actin were obtained from Santa Cruz Biotechnology (St. Louis, MO, USA). Anti-cyclin D1, anti-c-PARP, and anti-p-STAT3 (Tyr750) were obtained from Cell Signaling Technology (Beverly, MA, USA). Alexa 488-labeled goat anti-mouse IgG antibody was from Thermo Fisher Scientific, Waltham, MA, USA. ATD was provided by Dr. Lee and synthesized from 5-bromo-2-hydroxy-3-methoxybenzaldehyde, as previously reported [23]. Tris base and all other materials were purchased from Sigma Chemical Co. (St. Louis, MO, USA).

### 4.2. Cells and Cell Culture

The human liver cancer cell line Huh7 (*p53* Y220C) was obtained from the Food Industry Research and Development Institute (Hsinchu, Taiwan) and cultured in Dulbecco’s modified Eagle’s medium (DMEM) (Gibco BRL, Grand Island, NY, USA), supplemented with 10% FBS, 1% PSN, 1% essential amino acid, and 1mM glutamine. HepG2 (*p53* WT) cells were cultured in DMEM supplemented with 10% FBS, 1% PSN, 1% essential amino acid, 1% sodium pyruvate, and 1mM glutamine. PLC/PRF/5 (R249S) cells were cultured in MEM supplemented with 10% FBS and 1% PSN. The cell cultures were maintained at 37 °C in a humidified atmosphere of 5% CO_2_.

### 4.3. Cell Viability Assay

Huh7 and HepG2 cells were placed in a 24-well plate at a density of 2 × 10^4^ cells/well, respectively, and treated with various concentrations of ATD (10–80 μM) or solvent control (0.2% DMSO) for 24 h and 48 h. Cell viability was determined in the presence of 3-[4,5-dimethylthiazol-2-yl]-2,5-diphenyltetrazolium bromide (MTT) dye solution for 4 h. The medium was removed, and formazan was solubilized in isopropanol and measured spectrophotometrically at 560 nm using a microplate reader.

### 4.4. Trypan Blue Dye Exclusion Assay

Huh7 cells were placed in a 10 cm dish at a density of 4 × 10^4^ cells/dish and treated with ATD (10 μM) or solvent control (0.2% DMSO) for 24, 48, and 72 h. After treatment, trypan blue was added to the cell suspension, and viable cells that excluded the dye were counted on a hemacytometer.

### 4.5. Colony Formation Assay

Cells were plated in 6-well plates, at a density of 500 cells/well. On the next day, cells were treated with 0.2% DMSO (control) or ATD at the indicated concentration for 48 h, then cultured for 7 days. The colony was fixed with methanol for 15 min and stained with Giemsa. Cell colonies were photographed and counted.

### 4.6. Cell Cycle Analysis

Cell cycle distribution was determined using a flow cytometer with *propidium iodide* (PI) staining. Briefly, 6 × 10^5^ cells/dish were treated with 0.2% dimethyl sulfoxide (DMSO; control) or 10 μM ATD for indicated time. Then, cells were harvested, fixed with cold 75% alcohol, and stained with 50 μg/mL PI solution in darkness for 30 min on ice. The distribution of cells in different cell cycle phases was determined using flow cytometry (FACSCalibur, BD Biosciences, San Jose, CA, USA). In total, 10,000 cells per sample were counted, and DNA histograms were analyzed using Cell Quest software (BD Biosciences, San Jose, CA, USA) to calculate the percentage of cells in each peak.

### 4.7. Annexin V/PI Double Staining Assay

For this assay, 6 × 10^5^ cells were plated in a 10 cm culture dish. After attachment, cells were treated with DMSO or ATD at the indicated concentration for 48 h and then harvested and resuspended in PBS. Apoptotic cells were measured with a FITC-Annexin V Apoptosis Detection Kit (BD Biosciences, San Jose, CA, USA) according to the manufacturer’s protocol. Briefly, cells were stained with FITC annexin V and propidium iodide (PI) solution for 15 min at room temperature in darkness. In total, 10,000 cells were analyzed for each histogram. Flow cytometry demonstrated that the early apoptotic cells were in the lower-right quadrant, and the advanced apoptotic cells were in the upper-right quadrant. The apoptotic rate was the sum of the early and advanced apoptotic rates.

### 4.8. Western Immunoblotting

Equal amounts of protein from total cell lysates were separated in 8–12% polyacrylamide gel and transferred onto the PVDF membrane. The blot was subsequently incubated in blocking buffer (5% nonfat milk in PBS) for 1 h and then probed with a corresponding antibody against a specific protein overnight at 4 °C and washed with tris-buffered saline; the membrane was then incubated with an appropriate peroxidase-conjugated secondary antibody for 1 h. Finally, antigen–antibody complex was developed by ECL detection system. The relative image density was quantitated with densitometry.

### 4.9. Immunofluorescence

After ATD or DMSO treatment, Huh7, PLC/PRF/5, and HepG2 cells were washed with PBS and fixed with 4% paraformaldehyde for 10 min. The cells were permeated with 0.1% Triton X-100, then incubated at 4 °C overnight with a monoclonal anti-p53 (Pab-240) antibody, followed by a 1 h incubation with an Alexa 488-labeled goat anti-mouse IgG antibody (Thermo Fisher Scientific, Waltham, MA, USA). After washing with PBS containing 0.1% tween 20, the DAPI was added for 10 min. The cells were observed under a fluorescence microscope at 400× magnification.

### 4.10. Transfection with Small Interfering RNA (siRNA)

*p53* siRNAs (sense: 5′-AGA-CCU-AUG-GAA-ACU-ACU-Utt-3′) were purchased from GeneDireX, (QUANTUM BIOTECHNOLOGY, INC., Durham, NC, USA) [24]. For transfection, 3 × 10^3^ Huh7 cells were seeded on 96-well dishes or 4 × 10^5^ on 10 cm dishes. After overnight incubation, *p53* siRNA or control siRNA (40 nM) (Santa Cruz Biotechnology, Santa Cruz, CA, USA) were transfected using a T-Pro NTR II transfection reagent, according to the manufacturer’s instructions. Following incubation for 48 h, the cells were treated with or without ATD for 24 h. After ATD treatment, viable-cell counting was performed using Cell Counting Kit-8 (CCK-8 kit), or the total cell lysate was prepared for immunoblotting analysis.

### 4.11. Cell Proliferation Assay

Following the transfection, cell proliferation was assayed using a CCK-8, according to the manufacturer’s protocols. Briefly, after transfection and ATD treatment, the CCK-8 solution was added and incubated for 3 h. The optical density was measured at 450 nm using a microplate reader.

### 4.12. Statistical Analysis

Data are expressed as means ± SD from three independent experiments. The statistical significance of differences throughout the study was analyzed by a one-way ANOVA test. A *p* value < 0.05 was considered to be statistically significant.

## 5. Conclusions

This study demonstrated a novel mechanism in which ATD exhibited a more potent antiproliferation potential on mut*p53* HCC than on wt*p53* HCC cells due to the downregulation of mut*p53* and blockage of the STAT3 pathway (Figure 8).

## Figures and Tables

**Figure 1 ijms-23-05102-f001:**
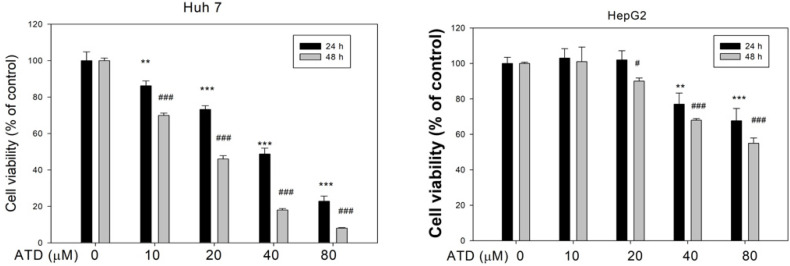
The effect of ailanthoidol (ATD) on the cell viability of Huh7 and HepG2 cells was determined with an MTT assay. Huh7 and HepG2 cells were treated with or without ATD under the indicated concentration for 24 h and 48 h. Cell viability was measured with an MTT assay, as described in the text. Data are represented as the means ± SD of three independent experiments. ** *p* < 0.01 and *** *p* < 0.001, compared with the control group (24 h) (0.2% DMSO) of the Huh 7 cells or HepG2 cells, respectively. # *p* < 0.05 and ### *p* < 0.001, compared with the control group (48 h) of the Huh 7 cells or HepG2 cells, respectively.

**Figure 2 ijms-23-05102-f002:**
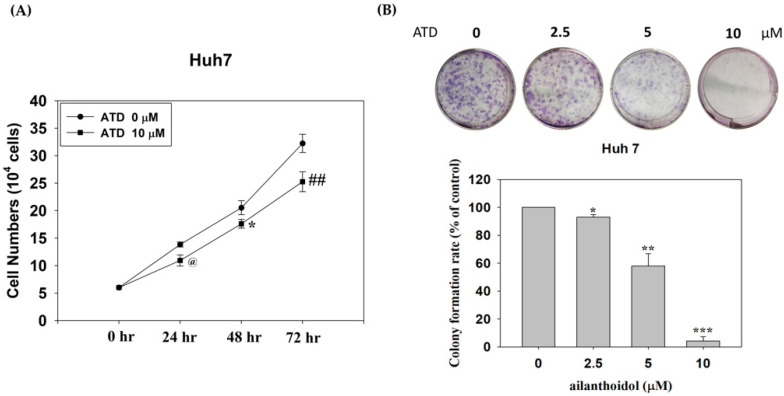
The effect of ailanthoidol (ATD) on the growth of Huh7 cells with a trypan dye exclusion assay and colony formation assay: (**A**) viable cells were counted using the trypan blue dye exclusion assay after treatment with ATD (10 μM) at 24 h, 48 h, and 72 h. Values are the means ± SD (*n* = 3). ^@^
*p* < 0.05, * *p* < 0.0 5, ^##^ *p* < 0.01, compared with the control group (0.2% DMSO) at 24 h, 48 h, and 72 h, respectively; (**B**) a total of 500 cells were seeded in a six-well dish. After attachment, the cells were treated with or without ATD (2.5, 5, and 10 μM) for 48 h and then cultured for seven days. The cells were fixed with methanol and stained with Giemsa. The number of colonies was counted. Data are represented as the means ± SD (*n* = 3). * *p* < 0.05, ** *p* < 0.01, *** *p* < 0.001, compared with the control group (0.2% DMSO).

**Figure 3 ijms-23-05102-f003:**
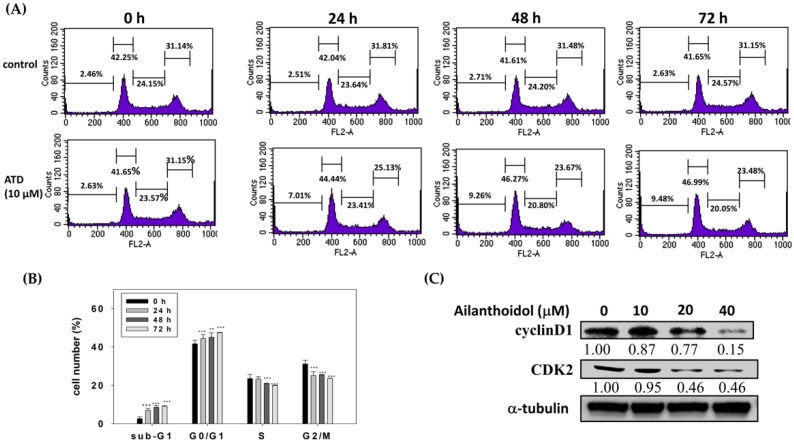
Ailanthoidol induced cell cycle arrest in Huh7 cells: (**A**) Huh7 cells were treated with or without ATD (10 μM) for 0, 24, 48, and 72 h. The harvested cells were stained with PI, and the DNA content was analyzed using a flow cytometer. The histograms are from one out of three experiments; (**B**) values are presented as means ± SD (*n* = 3). ** *p* < 0.01, *** *p* < 0.001 vs. o h; (**C**) Huh7 cells were treated with or without ATD for 24 h. The cells were harvested and equal protein amounts of the whole-cell extracts were analyzed by Western blotting against the indicated antibodies. α- tubulin was used as the loading control.

**Figure 4 ijms-23-05102-f004:**
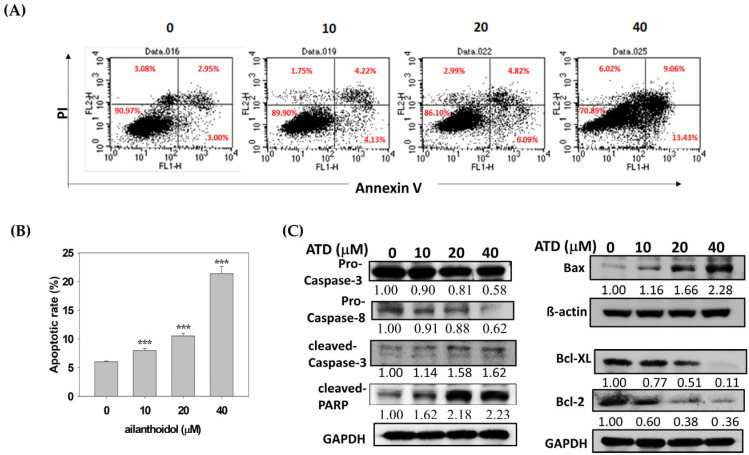
Ailanthoidol induced apoptosis in Huh7 cells. Huh7 cells were treated with or without ATD for 48 h: (**A**) the ATD-induced apoptosis in the Hun7 cells was determined by using a flow cytometer with Annexin V-FITC/PI staining, as described in the text. The cells in the lower-right quadrant (Annexin V+/PI-) represent the early apoptotic cells, and those in the upper-right quadrant (Annexin V+/PI+) represent the late apoptotic cells. A typical photograph from three independent experiments with similar results is shown; (**B**) data are presented as means ± SD (*n* = 3). *** *p* < 0.001 vs. the control; (**C**) Huh7 cells were treated with or without ATD for 24 h. The cells were harvested and equal protein amounts of the whole-cell extracts were analyzed with Western blotting against the indicated antibodies. β-actin or GADPH was used as the loading control.

**Figure 5 ijms-23-05102-f005:**
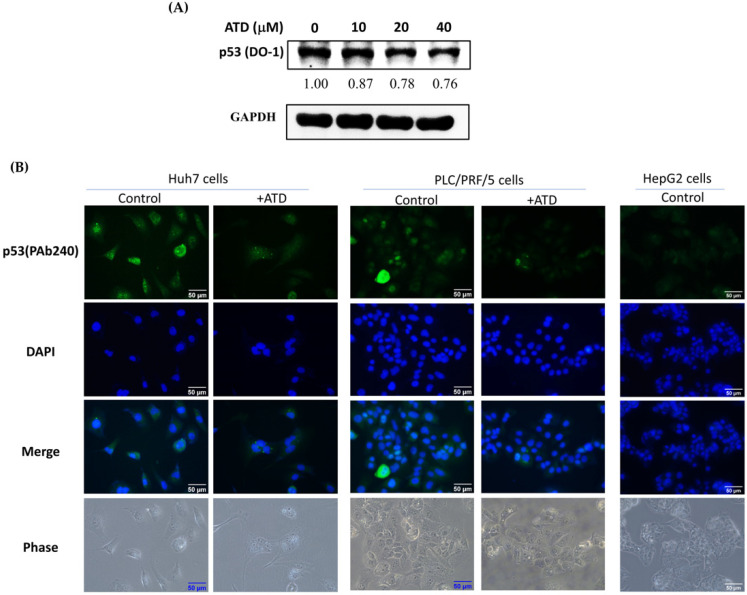
Mutant *p53* involvement in ATD-induced antiproliferation in Huh 7 cells: (**A**) after treatment with various concentrations of ATD for 24 h, the level of p53 in Huh7 cells was determined with immunoblotting analysis against anti-p53 (DO1). GADPH was used as loading control; (**B**) after treatment with DMSO (solvent control) or ATD (20 μM) for 24 h in Huh7 cells or PLC/PRF/5 cells, and HepG2 cells as negative control, the p53 against anti-p53 (PAb240) was detected with immunofluorescence analysis, as described in the text. The nuclear was stained by DAPI (blue). Green fluorescence indicated mut*p53*.

**Figure 6 ijms-23-05102-f006:**
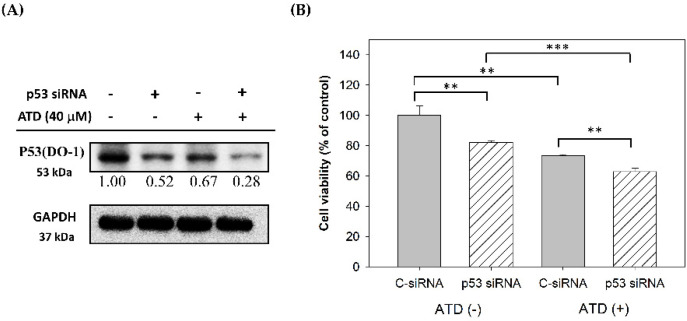
Antiproliferation effect of p53 siRNA transfection in Huh7 cells: (**A**) after 48 h of transfection with scrambled siRNA (C−siRNA) or siRNA against p53 (*p53* siRNA), Huh7 cells were treated with or without ATD (40 μM) for 24 h. The cell lysates were prepared, and the p53 levels were analyzed with Western blotting against p53; GADPH was used as the loading control; (**B**) after transfection and ATD treatment, the viable cells were determined by using a CCK−8 kit, as described in the text. Values are means ± SD (*n* = 3). The asterisks indicate statistic changes (** *p* < 0.01, *** *p* < 0.001).

**Figure 7 ijms-23-05102-f007:**
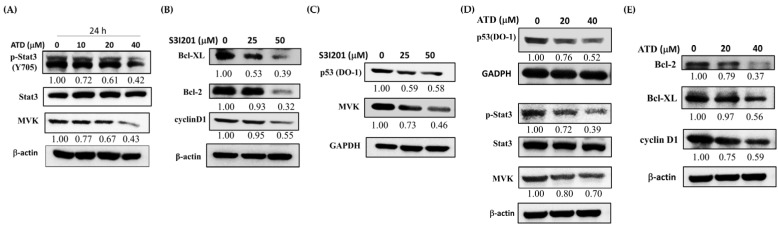
ATD-induced apoptosis and cell cycle arrest by blocking the STAT3 pathway in Huh7 cells: (**A**) after treatment with various concentrations of ATD for 24 h in Huh7 cells, the protein levels of p-STAT3, STAT3, and MVK were determined with immunoblotting. In addition, after treatment with various concentrations of S3I201 for 24 h in Huh7 cells, the protein levels of Bcl-XL, Bcl-2, and cyclin D1 (**B**), as well as p53 and MVK (**C**), were determined with immunoblotting analysis. β-actin or GADPH were used as the loading control. After treatment with various concentrations of ATD in the PLC/PRF/5 cells (mut*p53* R249S), the protein levels of p53 (DO-1), p-STAT3, STAT3, MVK (**D**), Bcl-2, Bcl-XL, and cyclin D1 (**E**) were determined with immunoblotting.

**Figure 8 ijms-23-05102-f008:**
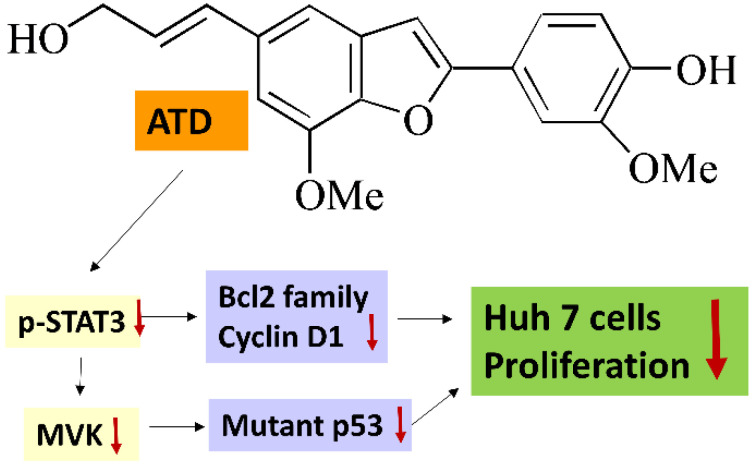
Antiproliferation of ailanthoidol (ATD) in mut*p53* hepatoma cells.

## Data Availability

All relevant data are within the paper.

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
