# Peer review of "Suppression of the Proliferation of Huh7 Hepatoma Cells Involving the Downregulation of Mutant p53 Protein and Inactivation of the STAT 3 Pathway with Ailanthoidol"

_ijms, 2022, doi:10.3390/ijms23095102_

Round 1

Reviewer 1 Report

The manuscript by Tsui-Hwa Tseng et al. describes the effect of Ailanthoidol (ATD) on Huh7 hepatoma cells, a cell line that harbors a mutate p53 protein. The authors evaluate ATD-mediated effects on cell proliferation, cell viability and induction of cell death by apoptosis and suggest that the antiproliferative and proapoptotic effect is mediated by the inhibition of STAT3 activation, that reducing the mevalonate kinase, down-regulates mutp53 expression. It is well known that STAT3 and mutp53 support proliferation of cancer cells, and it is plausible that their inhibition (by ATD) results in a decrease of proliferation and activation of cell death. The manuscript is well written, the bibliography is correct and the experiments well conducted.

However, I have a big concern about this work since the conclusions are based on experiments done with only one cell line. To confirm the authors’ conclusions it would be necessary to validate the experiments with at least one other mutp53 cell line. Moreover, it would be interesting to see what happens in terms of STAT3 and MVK activation in cells with wt p53.

Author Response

Apr 28, 2022

Associate Editor
IJMS Editorial Office

Manuscript ID: ijms-1693530 - Major Revisions

Dear Associate Editor,

Enclosed please find one revised version entitled: Suppression of the proliferation of Huh7 hepatoma cells involvingdown-regulating the mutant p53 protein and inactivating the STAT 3 pathway by ailanthoidol, which we would like to submit for publication in International Journal of Molecular Sciences.

This revised version has been carefully corrected according editor and referee’s reports point-by-point. We appreciate these valuable comments to strengthen our presentation. Please inform me if any revision is needed. The file marked change in blue color.

Furthermore, I would verify that no part of the manuscript is under consideration for publication elsewhere and it will not submit elsewhere if accepted by International Journal of Molecular Sciences and not before the Editorial Office has reached a decision.

Sincerely yours,

Hsing-Chun Kuo, Ph.D.

Professor

Department of Nursing,

Chang Gung University of Science Technology,

Chia-Yi Campus, Taiwan.                                                                                    E-mail: guscsi@gmail.com

TEL: +886-5-3628800

FAX: +886-5-3628866;

Revised version of ijms-1693530

Reviewer(s)' Comments to Author:
Reviewer #1: The manuscript by Tsui-Hwa Tseng et al. describes the effect of Ailanthoidol (ATD) on Huh7 hepatoma cells, a cell line that harbors a mutate p53 protein. The authors evaluate ATD-mediated effects on cell proliferation, cell viability and induction of cell death by apoptosis and suggest that the antiproliferative and proapoptotic effect is mediated by the inhibition of STAT3 activation, that reducing the mevalonate kinase, down-regulates mutp53 expression. It is well known that STAT3 and mutp53 support proliferation of cancer cells, and it is plausible that their inhibition (by ATD) results in a decrease of proliferation and activation of cell death. The manuscript is well written, the bibliography is correct and the experiments well conducted.

However, I have a big concern about this work since the conclusions are based on experiments done with only one cell line. To confirm the authors’ conclusions it would be necessary to validate the experiments with at least one other mutp53 cell line. Moreover, it would be interesting to see what happens in terms of STAT3 and MVK activation in cells with wt p53.

Response: We agree with this comment and have performed western blot assays in one other mutp53 cell line PLC/PRF/5 cells (mutp53 R249S) to assess the expression of p53 (DO-1), MVK and Bcl-2, Bcl-XL, cyclin D1 as well as and phosphorylation of Stat3 and quantitative analysis. We have described in the fig 7D and 7E. Moreover, we have shown that ATD significantly increased the percentage of total apoptotic cells  and reduced cell density and induced DNA fragmentation in replay 1.

The relationship of stat3, MVK, and wt p53 is interesting. According to the reference, it demonstrates that STAT3 inhibits the expression of w tp 53 at transcriptional level. Stat3 is a pro-oncogenic molecule while wtp53 acts as a tumor suppressor molecule whose function is incompatible with cancer development. The effect of ATD on stat3, MVK, and p53 in wt p53 cancer cells needs further clarified. (Ref.: Niu G, Wright KL, Ma Y, Wright GM, et al., Role of Stat3 in regulating p53 expression and function. Mol Cell Biol (2005) 25: 7432-7440.)

Replay 1

Reviewer 2 Report

The authors used a high concentration of ATD, can the authors justify using this conc. in in vitro setting is this clinically available?

No xenograft experiments were done to show the preclinical activity of ATD, can authors comment on this?

In Figure 4, cleaved caspase-3 was not shown

Authors show ATD decreases STAT-3 pathway, to show STAT pathway is involved, STAT-3 should be overexpressed and then show ATD activity is effected to prove STAT-3 is the mechanism of ATD induced apoptosis.

Author Response

Reviewer #2:.

The authors used a high concentration of ATD, can the authors justify using this conc. in in vitro setting is this clinically available?

Response: We appreciate this remark of the Reviewer:#2’s comment. Previous study has shown that the topical application of ailanthoidol  (0.5-2.5 mM; 200 μl) reduced the formation of hydrogen peroxide and inhibited the myeloperoxidase (MPO) activity in the mouse skin by 12-O-tetradecanoyl-phorbol13-acetate (TPA). We also investigated ATD’s selectivity against mutp53 hepatoma cells treated with ATD (10 μM). This dose has been used a low concentration of ATD to over 1/1000 fold, compared with in vivo mouse model. (Ref.: ONCOLOGY REPORTS 16: 921-927, 2006). In contrast, in vivo systems are like a vessel with a hole in it. They are open systems and then pharmacokinetics is the science of accurately determining the concentration of drug in the body and further studies will be needed to determine the devising methods of getting constant-state levels of ATD for clinically therapy through repeat dosing.

No xenograft experiments were done to show the preclinical activity of ATD, can authors comment on this?

Response: We agree with this comment. Although we found that ailanthoidol (ATD) reduced the level of MVK (the down-stream target gene product of STAT3, which might affect the stability of mutp53 in Huh 7 cells. Additional studies are still needed to elucidate the action mechanisms of the ailanthoidol (ATD) as a chemopreventive and therapeutic agent in vivo xenograft mouse model. We have described in the manuscript, page 10, lines 273 to 274, in the manuscript.

In Figure 4, cleaved caspase-3 was not shown

Response: We agree with this comment and have performed western blot assays to assess the expression of cleaved caspase-3 in Huh7 cells. We have described in the fig 4C.

Authors show ATD decreases STAT-3 pathway, to show STAT pathway is involved, STAT-3 should be overexpressed and then show ATD activity is effected to prove STAT-3 is the mechanism of ATD induced apoptosis.

Response: We appreciate this remark of the Reviewer:#2’s comment and have performed western blot assays in one other mutp53 cell line PLC/PRF/5 cells (mutp53 R249S) to assess the expression of p53 (DO-1), MVK and Bcl-2, Bcl-XL, cyclin D1 as well as and phosphorylation of Stat3 and quantitative analysis. ATD decreased the level of phosphorylated STAT3. For one other cell line PLC/PRF/5 cells (mutp53 R249S), ATD decreased phosphorylation of Stat3 (Figures 7A and 7D). The phosphorylation of Stat3 were significantly shown on both Huh7 hepatoma cells and PLC/PRF/5 cells. Further we are planning to establish stably and transiently expressing Stat3 hepatoma cells and valuate the mechanism of ATD involved in STAT-3 pathway.

We have revised in the fig 7D and 7E.

Round 2

Reviewer 2 Report

NA